# Antioxidant Potential of Spray- and Freeze-Dried Extract from Oregano Processing Wastes, Using an Optimized Ultrasound-Assisted Method

**DOI:** 10.3390/foods12132628

**Published:** 2023-07-07

**Authors:** Patroklos Vareltzis, Aggelos Stergiou, Kallirhoe Kalinderi, Maria Chamilaki

**Affiliations:** 1Laboratory of Food and Agricultural Industry Technologies, Chemical Engineering Department, Aristotle University of Thessaloniki, 54124 Thessaloniki, Greece; livonot1@gmail.com (A.S.); maria.chamilaki@hotmail.com (M.C.); 2Laboratory of Medical Biology-Genetics, School of Medicine, Faculty of Health Sciences, Aristotle University of Thessaloniki, 54124 Thessaloniki, Greece; kkalinde@auth.gr

**Keywords:** oregano, waste, ultrasound, antioxidant, spray drying, freeze drying, GAB

## Abstract

*Origanum vulgare* is recognized worldwide for its numerous applications, in the food industry and beyond. However, the extraction of its essential oils generates a significant amount of waste. The aim of this research was to achieve the valorization of solid waste from oregano hydro-distillation, by (i) optimizing the ultrasound extraction of antioxidants, (ii) evaluating the effect of spray and freeze drying on the extract’s physicochemical properties, and (iii) characterizing the obtained powder by its antioxidant capacity. A central composite design of experiments was used to optimize the sample/solvent ratio, ethanol/water ratio, and extraction time. The extract was analyzed for its antioxidant potential by determining the percentage of DPPH inhibition, FRAP, and total phenolic content (TPC). The GAB model best fit the data for the moisture sorption isotherm of the resulting powder. The antioxidant activity of the powders was tested in a ground-beef food system. The TPC was maximized at times longer than 58 min, a sample/solvent ratio between 0.058 and 0.078, and a ratio of ethanol/water around 1. Neither drying method significantly affected the antioxidant properties of the extract, even though the resulting powders from each showed a different morphology (determined using SEM). Encapsulation with maltodextrin protected the spray-dried extract during a 6-month storage period. Powders from both drying methods equally retarded lipid oxidation, and were comparable to the synthetic antioxidant BHT. It is concluded that oregano processing waste is a potent source of antioxidants, and that its dried extract, via an ultrasound-assisted process, can potentially be used as a natural alternative to synthetic antioxidants.

## 1. Introduction

Aromatic herbs are widely used in the food industry as additives, flavor enhancers, spices, aromas, and preservatives. Recently, they have also been used as promising raw materials to produce innovative products in the pharmaceutical, cosmetic, and insecticide industries. Rosmarinic acid, thymol, carvacrol, eugenol and other components of these herbs have demonstrated anticancer properties; the protection of the nervous system against oxidative stress; the ability to induct melanogenesis, and therefore be used in sun blocks; anticollagenase activity, useful in anti-aging products; and antiaflatoxigenic and antimicrobial activities [1,2,3,4]. “Greek oregano” (*Origanum vulgare* spp.) is recognized worldwide as the original and authentic oregano [5]. The essential oils of oregano account for more than 80% of its total composition, and carvacrol and thymol are the main volatiles in oregano, comprising 92% of the essential oil. Several studies conducted to date have shown that the strong antioxidant activity of oregano is due to these diterpenes [6].

There is a growing need in the food industry to replace synthetic antioxidants and preservatives with substances of natural origin. Serious concerns about synthetic antioxidants have been raised by various researchers. Attention should be given to the misuse or overuse of synthetic phenolic antioxidants, as they can lead to the development of cancer, harmful effects on cells, the induction of oxidative stress, and disruption of the endocrine system [7,8,9]. One of the most effective approaches to replacing synthetic antioxidants is the use of herbal extracts as a cost-effective alternative. Several studies have shown that herbs of the Lamiaceae family (such as rosemary, oregano, sage, basil, mint, and thyme) demonstrate great preservative, antimicrobial, and antioxidant activity, due to their high content of phenolic substances [10].

The exploitation of oregano is mainly based on the extraction of the essential oil, which is isolated from the plant material through the process of distillation. The steam and/or hydro-distillation methods are the most widespread ways of separating the oil from the plant material, but they generated significant amounts of waste, the utilization of which needs to be investigated. It is estimated that the volume of spent herb waste after oil extraction generally amounts to approximately 80% to 90% of the initial herb weight [11]. The valorization of the solid waste has been assigned as a top priority in many countries worldwide, and has the potential to serve as a valuable biomass [12]. It can be utilized in the extraction of valuable products, using modern chemistry and green extraction methods [13,14]. After the extraction of the essential oil, the rest of the plant material remains rich in phenolic compounds, flavonoids, and oligomers with a variety of bioactive properties; therefore, its valorization is of high importance [15].

The recovery of the antioxidant substances in the waste is carried out using an extraction method. Traditional extraction has been replaced by advanced methods, such as ultrasonic extraction, microwave extraction, and supercritical fluid extraction. Ultrasound extraction is more widely used in the food and pharmaceutical industry to extract bioactive substances from aromatic plants, and presents many advantages compared to traditional extraction, as it leads to a reduction in the time required, while achieving the same or even greater efficiency [16]. It also offers repeatability, and requires significantly smaller volumes of solvent, less energy, and lower temperatures [17]. The efficiency of ultrasound extraction can be influenced by various factors, including time, temperature, solvent type, and others. These factors can either positively or negatively impact the extraction process. The appropriate combination of these parameters gives the optimal performance of the process to isolate specific components. However, depending on the type of plant tissue being extracted, the optimization may differ [18].

On an industrial scale, substances in dried form are more desirable when it comes to reducing transportation costs. Moreover, drying is one of the oldest methods of food preservation, as it reduces water activity. Until now, different drying methods have been used for each type of food, such as spray drying and freeze drying. For plant antioxidants, such as polyphenols, encapsulation is a promising method of stabilization. Spray drying is one of the most commonly used encapsulation methods in the food and pharmaceutical industries, mainly because of its low cost [19]. For sensitive and thermo-liable ingredients in spray drying, maltodextrins (MD) are common encapsulants [20]. Freeze drying also has many advantages, but is considered the most expensive process for producing dried products [21].

In addition to conducting in vitro tests on the antioxidant potential of a bioactive compound, it is critical to evaluate its efficacy in real food systems. Lipid oxidation is a complex process, involving various factors, such as free radicals, inorganic polyvalent cations, oxygen, organic iron (e.g., heme-iron), and enzymes. The type, physical state, and concentration of lipids, as well as other food components like proteins, can influence this mechanism. Unlike in vitro systems, real foods contain a diverse range of inherent antioxidants, including both enzymatic (such as catalase and superoxide dismutase), and nonenzymatic components (such as ascorbic acid, tocopherols, and carotenoids) [22]. Furthermore, in the case of muscle foods, hemoglobin- and/or myoglobin-mediated lipid oxidation is one of the prevalent mechanisms for lipid oxidation, as hemoglobin and myoglobin act as potent pro-oxidants [23,24].

The objective of the present study was the valorization of solid waste from oregano processing, specifically the solid waste after hydrodistillation for oil recovery. To the best of our knowledge, there are few data in the literature on the utilization of distilled oregano solid wastes in recovering bioactive ingredients [25]. Therefore, the objectives of this work were: (i) to optimize the ultrasound extraction process, (ii) to evaluate the effects of different drying methods on the product, and (iii) to characterize the obtained powder in terms of its antioxidant capacity and physicochemical properties. Another important part of the study concerned the addition of these antioxidant powders to a real food matrix, to evaluate their antioxidant capacity in comparison with the synthetic antioxidant butylated hydroxytoluene (BHT).

## 2. Materials and Methods

Dried Greek oregano (*Origanum vulgare* ssp. hirtum) was purchased from a local market, “Kapani”, in Thessaloniki, Greece. Fillets of cod (*Gadus morhua*) were purchased from a local vendor (Thessaloniki, Greece). Lean ground beef was purchased from a local butcher’s shop in Thessaloniki, Greece. All chemicals, reagents, and solvents were of analytical grade, and were obtained from SIGMA Chemical Co., St. Louis, MO, USA.

### 2.1. Preparation of Distilled Oregano Waste

To produce the distilled oregano waste, and the separation of volatile and non-volatile substances, hydro distillation was carried out, using a Clevenger-type apparatus. The duration of the distillation was 2 h, as the oil yield remained constant after 80 min [26]. The waste from the distillation process was filtered to remove the excess water, placed in an oven (MELAG Apparate GmbH) at a temperature of 40 ± 2 °C for drying, and stopped when the moisture content reached 14%. Preliminary experiments in our lab showed that this moisture percentage seemed to be the most suitable value for the extraction of phenolics. Finally, the dried raw material was milled in a knife mill with a 500 μm sieve (Polymyx, PX-MFC 90D, Kinematica AG, Malters, Switzerland).

### 2.2. Ultrasound Extraction Optimization

Various parameters can affect ultrasound extraction efficiency, such as the temperature, solvent ratio (S/S), type of solvent, sample pretreatment, and extraction time, as well as the ultrasonic power/amplitude. The parameters to be optimized, as well as the relative value ranges, were selected after preliminary measurements, in conjunction with data from the literature [16,27].

To optimize the extraction process, a central composite design (CCD) with a two-level full factorial approach was selected, where α was set to ±1.68. The use of a central composite design enables the assessment of nonlinear responses within the provided dataset. It also aids in estimating the curvature of continuous responses, while ensuring maximum information is obtained with a minimal number of experimental trials. Additionally, this design reduces the number of trials needed to estimate the squared terms in the second-order model [28].

For the extraction of phenolic compounds, a SONOREXTM Digital 10 P Ultrasonic Bath (Bandelin Electronic GmbH & Co. KG, Berlin, Germany) was used. The experimental design and the obtained data were analyzed using the statistical software Minitab (© 2023 Minitab, LLC, State College, PA, USA). A two-level full factorial central composite design was chosen, and 20 experiments were conducted. The three selected factors were the sample-solvent ratio (S/S), ethanol-water ratio (E/W), and extraction time. The DoE is presented in Table A1 in Appendix B. All extractions were performed at a temperature of 35 ± 2 °C, and the ultrasound power was set at 50% of the maximum power (180 W). The optimal conditions determined by the statistical analysis were used for the rest of the experiments.

### 2.3. Determination of the Total Polyphenolic Content (TPC)

The total amount of phenolic compounds was determined spectrophotometrically, using the Folin–Ciocalteu (FC) phenol reagent, according to Filippou et al. (2022) [29]. The absorbance was measured at 760 nm, using a Thermo Spectronic Helios Gamma spectrophotometer. The total phenolics were expressed as gallic acid equivalents (g GAE/100 g solid oregano waste). All experiments were performed in triplicate.

### 2.4. Determination of %Antioxidant Capacity

The determination of the radical scavenging activity was calculated by the decolorization of the DPPH solution, according to Filippou et al. (2022) [29]. The antioxidant capacity was typically calculated as a percentage, using the following Equation (1):(1)$$Andioxidant\;Capacity\;(\%)=\frac{A_{DPPH}-A_{S}}{A_{DPPH}}$$
where A_DPPH_ is the absorbance of the blank sample, and A_s_ is the absorbance of the sample.

In the storage stability study, the antioxidant capacity was expressed as EC_50_. The EC_50_ value expresses the concentration of the antioxidant required to bind free radicals to decrease 50% of the DPPH absorbance.

### 2.5. Determination of the Ferric Reducing Antioxidant Power (FRAP Assay)

A phosphate buffer of 0.2 M and pH 6.6 was initially prepared. Then, 2 mL of water, 2.5 mL of 1% *w*/*v* potassium ferricyanide solution, and 2.5 mL of buffer were added to a test tube. This mixture was placed in a hot bath at 55 °C, for 30 min. Afterward, 2.5 mL of 10% *w*/*v* trichloroacetic acid solution was added, and the solution was centrifuged at 2000 rpm for 10 min. Finally, 2.5 mL of the supernatant liquid of the solution, 2.5 mL of water, and 0.5 mL of 0.01% *w*/*v* ferric chloride solution were mixed. The absorbance of this solution was measured at 700 nm. The molar absorptivity (ε) value for TPTZ was set to 22,230, according to data in the literature [30]. The sample concentration was calculated in µmol equivalents TPTZ/L, or in mg ascorbic acid equivalents (AAE)/g dry base of waste.

### 2.6. Spray and Freeze Drying

The extract was concentrated 10 times, using a rotary evaporator (Laborota 4003, Heidolph Instruments GmbH & Co. KG, Schwabach, Germany), at a temperature of 40 ± 2 °C, before spray or freeze drying to remove most of the organic solvent, and to increase the TDS of the solution to 10%.

The spray drying was carried out in a spray dryer Buchi-191 (BÜCHI Labortechnik AG, Flawil, Switzerland). The inlet air temperature was 170 °C, the pump power was set to 5% of the maximum, the drying air rate was set to 0.27 m^3^/h, and the atomizing air pressure to 3 bars [31]. These conditions led to an outlet temperature between 55 and 60 °C in all cases. For the encapsulated powder, maltodextrin was chosen as the encapsulating agent. Maltodextrin (DE10) was added at a final concentration of 10% *w*/*w*. The final dried powder was collected from the cyclone, packed in a vacuum, and stored under cold conditions (5 °C) until its analysis.

For the freeze drying, the concentrated extract was frozen at −18 °C for 48 h, and then placed in an Alpha 1–2 freeze dryer equipped with a capacitance manometer, to monitor the condenser pressure (Martin Christ Gefriertrocknungsanlagen GmbH, Osterode am Harz, Germany) for another 48 h under vacuum (1.6 mm Hg), until a constant moisture content was achieved. The chamber and ice condenser temperatures were 20 °C and −55 °C, respectively [32]. The powder was stored under refrigeration for future use.

### 2.7. Moisture Sorption Isotherm

The isotherm curves were calculated at temperatures of 20 °C, 30 °C, and 40 °C. To maximize the surface area, 3 g of powder was spread evenly in a pre-weighed Petri dish. The Petri dishes were placed in desiccators with specific relative humidity (RH) and water activity (α_w_). Sulfuric acid solutions were used for the adjustment of these conditions. The samples were allowed to equilibrate until there was no noticeable weight change (±0.001 g). It took approximately 40 s to remove and weigh the samples, and return them to the desiccators. This ensured that minimal moisture was absorbed from the surroundings while weighing the samples. The equilibrium moisture content was determined by drying in an oven at 70 °C, until constant weight was achieved [33]. All samples were prepared in triplicate. The results were compared with theoretical models (GAB and BET), empirical models (Oswin and Peleg), and Smith’s semi-empirical models (Table 1).

The model parameters were determined using non-linear regression. The least-square method was used to find the best fit of data, and to minimize the average standard deviation Mc (%), which is expressed with the following equation:(2)$$M_{c}=\frac{100}{N}\;\sum_{i=1}^{N}\frac{\left|m_{i-}m_{pi}\right|}{m_{i}}$$
where m_i_ is the experimental value of the measurement, m_pi_ is the predicted value, and N is the number of experimental values. The determination of the constants of the GAB equation (Table 1) can be achieved in two ways, the direct and the indirect method. In this study, the coefficients of the mathematical model were determined by the direct method (see Appendix B for details).

### 2.8. Scanning Electron Microscopy of the Powder (SEM)

Scanning electron microscopy (SEM) was used to determine the particle morphologies of the antioxidant powders. The dried samples were attached to an adhesive tape on SEM stubs, and coated with gold. The operation was performed at 20 kV, and the images were captured at 100–3000× magnifications. The measurements were conducted at the Center for Research and Technology Hellas (CERTH, Thermi, Greece).

### 2.9. Determination of the Oxidation Rate in a Ground-Beef Matrix by Thiobarbituric Acid Reactive Substances (TBARSs)

A chopped extra-lean cut of beef was chosen as the matrix. The fat quota of this specific tissue was ≈4.5% and the polyunsaturated fatty acids (PUFA) were ≈4.9% of the total fats. Six different samples were prepared, consisting of 40 g of ground beef and 10 mL of water. A different antioxidant was dissolved in that 10 mL of distilled water: 500 ppm of encapsulated, non-encapsulated, freeze-dried powders, and 100 ppm of BHT and 100 ppm maltodextrin. The blank sample did not contain any additives in the added water. After being thoroughly mixed with a spatula, the samples were evenly distributed in Petri dishes, covered with aluminum foil, and stored in the refrigerator (5 °C). Each sample was prepared in triplicate. To determine the secondary oxidation products, the TBARSs were determined. A sample of 1.5 g minced meat was homogenized with 5 mL of 7.5% *w*/*v* trichloroacetic acid (TCA) solution. The mixture was centrifuged for 30 min at 4000 rpm. Then, 2 mL of the supernatant was added to 2 mL of 0.02 M thiobarbituric acid (TBA) solution. The mixture was placed in a water bath, at a temperature of 100 °C, for 40 min. The absorbance was measured spectrophotometrically, at a wavelength of 532 nm. A 1:1 ratio of TBA and TCA solution was used as the blank. The standard curve was constructed using known concentrations of tetraethoxypropane (TEP). The results of the measurements are expressed in μmol MDA/kg tissue.

### 2.10. Statistical Analysis

The statistical analysis was performed using the SPSS software for Windows, version 25 (IBM SPSS Statistics 25). The level of statistical significance (*p*-value) was set at 5%. A *p*-value less than 0.05 was considered statistically significant. To study the significance of the effects and interactions between the response variables, a one-way analysis of variance (ANOVA) test was performed for each, and Tuckey’s post-hoc test was carried out for means comparison.

## 3. Results

### 3.1. Ultrasound Extraction Optimization

After the preliminary experiments, and taking into consideration data in the literature, we chose the optimizing parameters for increasing the antioxidant capacity of the extract to be the sample-solvent ratio (S/S), ethanol-water ratio (E/W), and extraction time. The response variables were the total phenolic content (TPC) determined using the Folin–Ciocalteu method, the ferric-iron-reducing capacity using the FRAP method, and the percentage of antioxidant capacity using the DPPH method. The extract content in the TPC varied from 1.6 to 5.8%. The antioxidant capacity ranged from 77% to 88%. The FRAP method gave TPTZ concentrations from 2.5 up to 11.9 μmoles/L. The detailed results are presented in Table A2 of Appendix B.

The ANOVA results (Appendix A) and the response surface plots (Figure 1) indicate that time was the most important factor in all three response variables. In fact, it can be observed that there was a strong linear dependence of FRAP, TPC, and the percentage of antioxidant activity with time. The FRAP and TPC increased with the extraction time, while the percentage of antioxidant capacity linearly decreased with time, with a total decrease in 70 min of around 10%. The FRAP showed a strong dependence (*p* < 0.05), and the TPC a weak dependence (*p* = 0.067), on the ratio E/W. In Figure 1a–c, the effect of the three studied parameters on each response variable can be observed. In Figure 1a,b, the significant effect of the quadratic terms of the regression analysis is observed, which was also verified by the ANOVA analysis (Appendix A).

To determine the optimum range of conditions for maximum TPC extraction, contour plots were constructed. In Figure 2, it can be seen that the TPC was maximized at times longer than 58 min, and a sample-to-solvent ratio between 0.058 and 0.078, while a ratio of ethanol to water of around 1 seemed to be sufficient to achieve maximum TPC recovery in the extract.

### 3.2. Effect of Drying and Encapsulation on the Extract’s Antioxidant Properties and Stability

The extract was concentrated, until its solids content (TDS) equaled 10%. The spray-drying solid yield was found to be approximately 40%. For the encapsulated antioxidant powder, the yield was found to be 55%.

For the TPC, the percentage of antioxidant activity, measured as the EC_50_ and FRAP, was determined before and after spray or freeze drying, to determine whether the drying method had any negative effect on the antioxidant properties of the extract. Then, to test the storage stability of the powders, the samples were packed under a vacuum, stored under refrigerating conditions for six months, and then analyzed again. The results show that neither spray nor freeze drying significantly affected any of the measured antioxidant properties (*p* > 0.05). Encapsulation with maltodextrin slightly decreased TPC and FRAP. On the other hand, encapsulation significantly increased the stability of the powder during storage, in terms of the TPC and FRAP (Table 2).

### 3.3. Particle Morphology

The surface morphologies of the antioxidant powders were examined using scanning electron microscopy (SEM), and are presented in Figure 3.

Differences between the dying methods were noticed. When the external morphology of the particles was observed, most of them showed a variety of sizes, with diameters in micrometer scale. This picture depended on the nature of the material, and the method used for its preparation. The particles obtained by spray drying seemed to have a spherical shape, with some cavities on their surface (Figure 3a,b).

### 3.4. Sorption Isotherms

The Peleg model was the best fit for the data, with the lowest average standard deviation Mc (%) (Table 3). Nevertheless, the theoretical model GAB was chosen to generate the isotherm curves (Figure 4). The theoretical mathematical model GAB adequately described the trend of the moisture content of the samples to increase with the water activity, with the relative percentage errors not exceeding 6%. In addition, the coefficients of the model, Xm, C and K, have a physical meaning, and can provide important information about the properties of each powder under different storage conditions. The above coefficients were calculated using the direct regression method. The Xm coefficient ranged from 5.78 to 8.5% for non-encapsulated spray powder, 4.86 to 6.27% for encapsulated powder, and 3.85 to 5.04% for freeze-dried powder (Table 3).

The values of the model GAB were also used to calculate the water-activity values of each powder. The results for the different temperatures during the storage time are presented in Table 4. The different temperatures were chosen according to the minimum degradation of the antioxidant powders. At higher temperatures, other reactions are possible, because the moisture content of each powder in equilibrium exceeds that of the monolayer.

Taking into account the values of the parameters of the GAB model, we designed Table 4, which presents the values of water activity during the storage of each powder at different temperatures, below which the minimum degradation was observed. As the humidity of each powder in equilibrium exceeded that of the monolayer, additional reactions became possible.

### 3.5. Food Matrix

To investigate the antioxidant activity of different antioxidants, the control sample (i.e., without antioxidant) and the different powders obtained by spray and freeze drying were compared with the antioxidant activity of BHT (Figure 5). Maltodextrin was also tested for its potential to retard lipid oxidation, as it was used as an encapsulating agent. It showed no protection against oxidation. The rate of oxidation appeared to have decreased significantly in all other samples, compared to the control and maltodextrin (*p* < 0.05). The freeze-dried powder, however, showed stronger antioxidant activity, compared to the spray-dried ones, on the fourth and fifth day of storage (*p* < 0.05), showing a similar behavior to BHT (*p* > 0.05). No significant difference in the retardation of lipid oxidation was observed between the encapsulated and non-encapsulated powders (*p* < 0.05).

## 4. Discussion

Several studies utilizing distillation, ultrasound, microwave, or supercritical extraction techniques have highlighted the rich composition of bioactive ingredients present in oregano, especially in its oil, and the potential for its application in the food and pharmaceutical industries [34,35,36,37,38]. The waste that is produced from oregano primary processing can be equally rich in bioactive ingredients [28].

At the optimum range of conditions for extraction (time > 58 min, E/W ratio equal to 1, and S/S ratio 0.058–0.76), the results for the TPC, percentage of antioxidant activity, and FRAP were very promising. The antioxidant activity measured in mgAAE/g dry basis was higher than the corresponding activity of Mexican oregano waste, where the extraction was performed with ethanol and ethyl acetate (26 and 12 mg AAE/g dry base, respectively) [39]. This value indicates the significant antioxidant activity of oregano, even after its oil was removed. The TPC was comparable to, or slightly less than, other studies in the literature, where an ultrasound-assisted or pressurized liquid extraction (PLE) methodology was applied to extract phenolics from various types of untreated oregano [40,41]. This indicates that during the hydro-distillation, most of the phenolic compounds remained in the wastes and, possibly due to the mass and heat transfer phenomena, certain phenols became more available for the subsequent ultrasound-assisted extraction. Other parameters that could be optimized, to further improve the recovery yield for TPC and antioxidant activity, are the temperature, and the particle size of the dried raw material [42].

The results show that the optimal extraction time was between 58 and 70 min. According to the literature, an increase in time consistently leads to an increase in yield in TPC and FRAP, but with a decrease in the extraction rate [43]. The optimal ethanol–water ratio was 1:1, which agrees with the literature [27]. It is even reported that, with an increase in the amount of organic solvent compared to water beyond the ratio of 70/30, the yield decreases significantly; meanwhile, using a small amount of organic solvent, the yield is equally small. Finally, the optimal sample–solvent ratio was in the range of 1 g/13 mL to 1 g/18 mL, while the literature reports ratios from 1 g/10 mL up to 1 g/40 mL for the extraction of aromatic plants [44,45].

Spray drying the extract without maltodextrin as an encapsulation agent slightly decreased (*p* > 0.05) the TPC and FRAP, while freeze drying had no effect. On the other hand, the use of maltodextrin for spray drying decreased the TPC and FRAP. The effect of spray drying on the TPC and antioxidant power in general is affected mainly by inlet temperature, and the concentration of the drying agent, maltodextrin in this case. An increase in maltodextrin from 5 to 9% showed a significantly decrease in TPC in samples, a phenomenon attributed to maltodextrin’s concentration effect [46]. Other researchers have observed a strong dependence of the total antioxidant activity on the inlet temperature and maltodextrin concentration when spray drying gac juice [47]. The optimization of spray-drying conditions, considering the inlet temperature, and the type of drying agent and its concentration, is deemed necessary, to improve the stability of TPC during drying. There are also references in the literature suggesting that spray drying can cause a decrease in the total phenolic content and antioxidant activity of powders, suggesting a potential loss or modification of phenols during the drying process [48,49,50]. However, it is important to note that the impact of spray drying on phenolic activity can vary depending on various factors, such as the drying conditions, initial phenolic content, and specific plant materials. Encapsulation with maltodextrin, on the other hand, seemed to protect the phenolic compounds and the ferric-reducing antioxidant power during storage to a significant degree. Maltodextrin is a polysaccharide obtained from starch using acid hydrolysis, and it is widely used in the food industry as a drying and/or encapsulation agent in spray drying [51]. Maltodextrin, as an encapsulation agent, can form a protective matrix around phenols, reducing their exposure to harsh processing conditions, and preventing their degradation [52].

For the encapsulated antioxidant powder, the efficiency in the solid yield was found to be 55%, which agrees with the literature [53]. The higher efficiency of the encapsulated antioxidant powder was attributed to the increase in the density of the solution, as well as the increase in Tg and, as a result, the deposition of dust on the walls of the drying chamber was lesser.

The distinct morphology of the spray-dried particles was in agreement with other studies, in which hollow particles were also produced by spray drying [54,55]. As stated, some of these cavities may be attributed to the shrinkage process, when the integrated bubbles in the drop dilated. Another explanation for the emerged cavities may be air bubbles entrained through the atomization [56]. As for the external morphology of the encapsulated powder, it presented a porous surface structure, and some aggregates on the surface. Under that porous structure, the bioactive compounds could be entrapped or adhered to [57]. On the other hand, the lyophilized powder presented a crystal structure with sharp edges (Figure 3c).

The calculated parameter values of the GAB model agreed with earlier research on dried extracts [58]. This value referred to the moisture content of the monolayer, and it was recognized as the moisture at which the least quality degradation appeared in the material. Below this moisture value, degradation reactions were minimized. The oxidation of unsaturated fats was an exception. Consequently, each material showed a critical value of moisture, which should be determined during storage.

Isothermal moisture sorption curves are now available for several foods, as well as for several aromatic plants, but no data were found for the specific powders in our study. Furthermore, the change in the value of activity as a function of temperature is often not considered, resulting in errors in the calculations [59]. The influence of temperature on isothermal sorption is of great significance, especially considering that food products are subjected to different temperatures during storage and processing. These temperature variations can lead to substantial alterations in the water activity, even at the same moisture content. Temperature directly impacts the movement of water molecules, and the balance between vapor and adsorbed moisture [60]. Experimental evidence suggests that, at a constant water activity, the equilibrium moisture content decreases as the temperature increases. This observation can be attributed to a reduced number of available sites where water can bind, thereby causing physical or chemical transformations.

At higher temperatures, water molecules are activated at higher energy levels, escaping from the positions where they were bound in the food, leading to a reduction in the moisture content. However, many researchers have reported that a high content of sugar, such as glucose, could have a different effect on the temperature at higher water-activity levels. In the literature, it has been mentioned that the moisture content in raisins, at water-activity levels below 0.55, decreases with an increasing temperature, while at higher water-activity levels, an increase in temperature leads to an increase in the moisture content [61]. This phenomenon seems to explain the trend of encapsulated powders with high water-activity values (>0.8) seen in Figure 4b.

Spray- or freeze-dried extracts from oregano waste, with or without maltodextrin encapsulation, were successful in retarding lipid oxidation in beef patties. Dried extracts from various plants or plant-processing wastes have been shown to possess antioxidant activity when tested in real food matrices, such as meat and poultry [62,63]. The fact that encapsulation did not further improve antioxidant activity is corroborated by the findings in Table 2, where there was no significant difference in the TPC, antioxidant capacity, and FRAP between the encapsulated and non-encapsulated spray-dried extract. The freeze-dried extract, on the other hand, with higher TPC and FRAP values than the spray-dried extracts, exhibited better antioxidant activity, comparable to that of BHT. A spray-dried concentrate from wine lees was also potent in retarding lipid oxidation in a similar food system [29]. When it comes to comparing the efficiency of freeze and spray drying in retaining antioxidant potential, it is of paramount importance to carefully select the encapsulation agent, and to optimize the operating conditions of the drying technique, for a successful incorporation and retention of bioactive compounds [64,65].

## 5. Conclusions

This study concludes that solid wastes from oregano hydro-distillation are rich in antioxidant potential. Therefore, their valorization is of great importance. In this research work, we propose an optimized ultrasound extraction method for the recovery of phenols and other antioxidants, followed by spray or freeze drying. Among the studied parameters for the extraction, time seemed to be the most important factor affecting the TPC, percentage of antioxidant activity, and FRAP. A sample-to-solvent ratio between 0.058 and 0.078, and an ethanol-to-water ratio around 1 produced the optimum results. The dried extracts retained most of their antioxidant properties, while maltodextrin helped to maintain these properties over a period of 6 months. Distinct variations in the powders’ morphology and isothermal moisture sorption behavior were observed due to different drying techniques and the utilization of maltodextrin as an encapsulation agent. These findings strongly suggest that spray-dried and freeze-dried extracts from oregano hydrostillation processing wastes hold promise as natural alternatives to synthetic antioxidants for preventing lipid oxidation in the food-processing industry. It is also evident that encapsulation is vital for enhancing the storage stability of the dried extracts. Further research is necessary, to explore the bioactivity of these extracts, including their behavior during gastrointestinal digestion, bioavailability, potential interactions with other food components, and toxicity. Furthermore, a comprehensive profile analysis of the bioactive compounds present in these extracts should be conducted via analytical techniques, i.e., LCMS.

## Figures and Tables

**Figure 1 foods-12-02628-f001:**
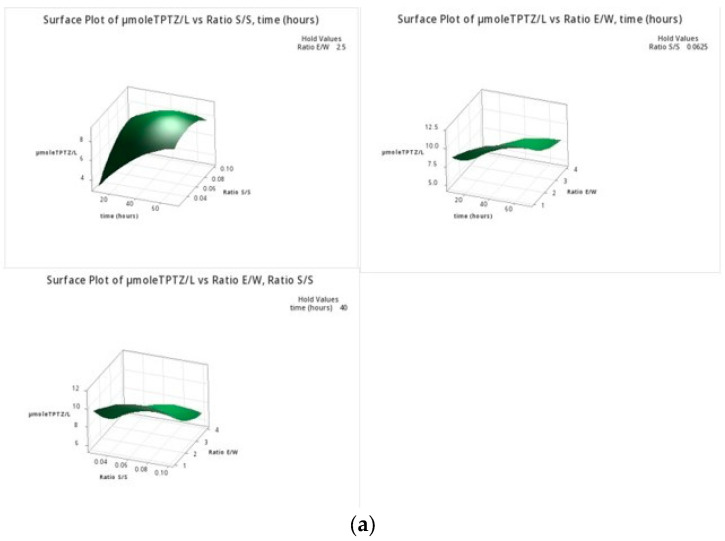
Surface plots of the effect of the E/W ratio, S/S ratio, and time (min) on (**a**) FRAP in μmoles TPTZ/L, (**b**) percentage of antioxidant capacity (%DPPH), and (**c**) TPC in g GAE/g of waste. All surface plots were constructed at the median hold values of the parameters (time in min, ratio S/S, and ratio E/W).

**Figure 2 foods-12-02628-f002:**
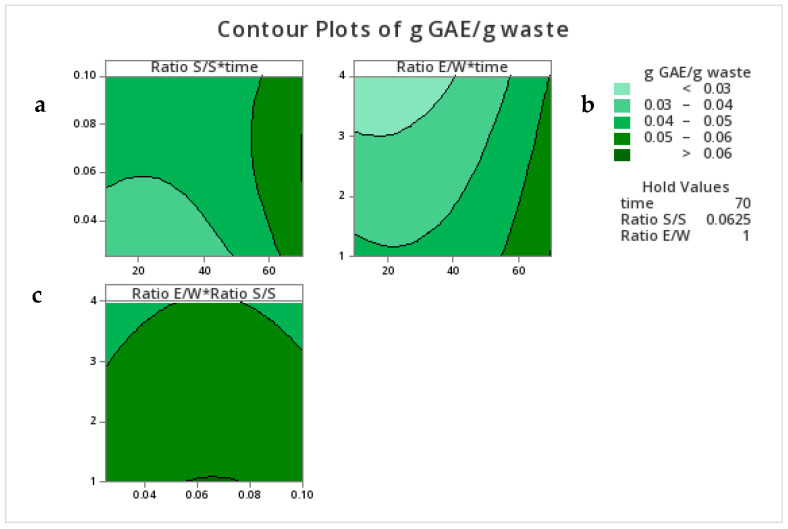
Contour plots of TPC (g GAE/g waste). (**a**) TPC as affected by the ratio S/S (*Y*-axis) and time in min (X-axis), (**b**) TPC as affected by the ratio E/W (*Y*-axis) and time in min (X-axis), and (**c**) TPC as affected by the ratio E/W (*Y*-axis) and the ratio S/S (X-axis).

**Figure 3 foods-12-02628-f003:**
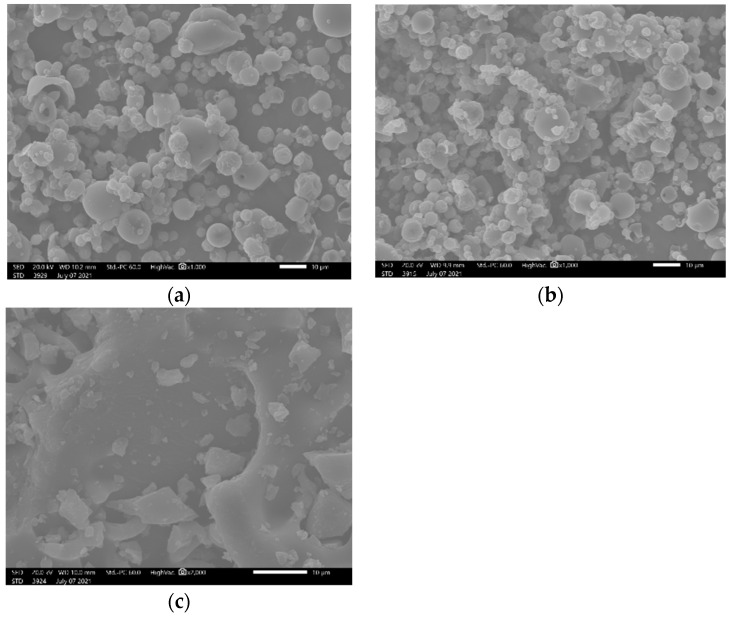
Surface morphologies of (**a**) non-encapsulated powder, (**b**) encapsulated powder, and (**c**) freeze-dried powder of distilled oregano waste, with a magnification of 1000×.

**Figure 4 foods-12-02628-f004:**
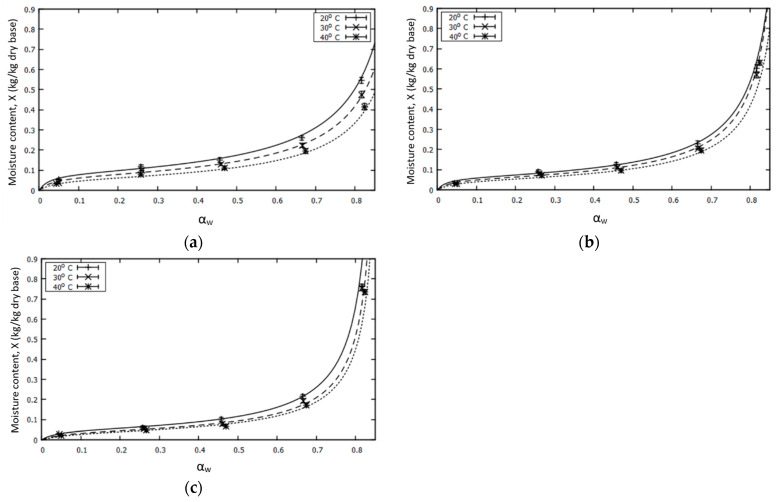
Sorption isotherms of (**a**) non-encapsulated spray-dried powder, (**b**) encapsulated spray-dried powder, and (**c**) freeze-dried powder, at different temperatures, and predicted lines using the mathematical model, GAB.

**Figure 5 foods-12-02628-f005:**
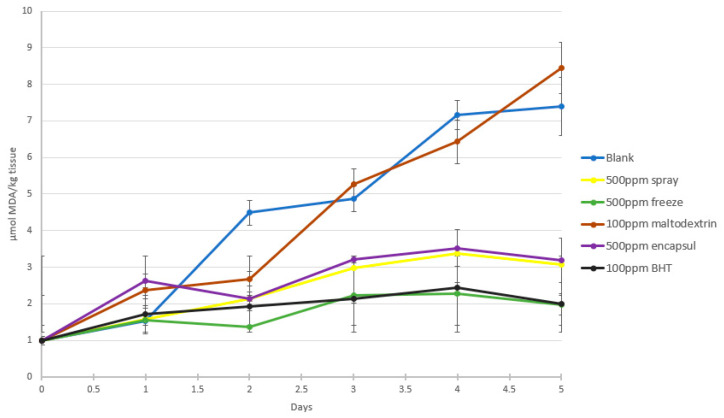
Effect of different antioxidant powders on ground-beef lipid oxidation.

**Table 1 foods-12-02628-t001:** Description of different mathematical models of sorption.

Model	Mathematical Expression	Constants
BET	*X* = $\frac{XmCaw}{\left(1-aw)(1-aw+Caw\right)}$	X_m_, C
GAB	*X* = $\frac{XmCKaw}{\left(1-Kaw)(1-Kaw+CKaw\right)}$	X_m_, C, K
Smith	*X* = *A + Blog* (1 − *α_w_*)^1/*B*^	A, B
Oswin	*X* = *A* [*a_w_*/(1 − *α_w_*)] *^B^*	A, B
Peleg	*X* = *K*_1_*a_w_^n^*^1^ + *K*_2_*a_w_^n^*^2^	n_1_, n_2_, K_1_, K_2_

Where: X, moisture content (% dry basis); Xm, monolayer moisture content (% dry basis); T, temperature (K); C, K, A, B, K_1_, K_2_, n_1_, n_2_, constants.

**Table 2 foods-12-02628-t002:** Comparison of different methods of measuring the antioxidant capacity and phenolic content of the samples.

Antioxidant Sample	Total Phenolic Content (GAE mg/g Dry Base)		Free Radical Absorption (EC_50_) μg/mL		FRAP (AAE mg/g Dry Base)	
	Initial Values	6-Month Interval	Initial Values	6-Month Interval	Initial Values	6-Month Interval
**Extract**	367.3 ± 11.5 ^a^		287 ± 1.2 ^a^		691.5 ± 9.2 ^a^	
**Non encapsulated powder**	341.4 ± 11.2 ^ab^	201 ± 15.2 ^a^	292 ± 0.1 ^a^	247.3 ± 9.1 ^a^	625.4 ± 12.3 ^ab^	244.5 ± 6.1 ^a^
**Encapsulated powder**	312.7 ± 18.2 ^bc^	256.6 ± 12.2 ^b^	228 ± 1.2 ^ab^	231.2 ± 6.0 ^a^	581.1 ± 18.2 ^b^	382.6 ± 12.6 ^b^
**Freeze-dried powder**	368.2 ± 19.0 ^a^	268.5 ± 19.1 ^b^	296 ± 1.5 ^a^	241.4 ± 3.3 ^a^	675.0 ± 19.0 ^a^	322.6 ± 11.0 ^c^

No statistically significant differences between the values in the same column with the same letter (*p* > 0.05).

**Table 3 foods-12-02628-t003:** Parameters of the mathematical models for the different antioxidant powders.

Model	Constant	Temperature (°C)								
		Non-Encapsulated Powder			Encapsulated Powder			Freeze-Dried Powder		
		20	30	40	20	30	40	20	30	40
**GAB**	X_m_ (dry mass)	8.56	7.11	5.70	6.27	5.55	4.88	5.04	4.14	3.83
	C	36.7	27.98	21.67	39.10	28.44	21.11	22.58	16.57	12.40
	K	1.04	1.039	1.039	1.112	1.108	1.105	1.155	1.150	1.144
	M_c_ (%)	4.59	6.89	6.08	9.80	5.38	8.56	4.71	12.97	10.36
	AverageM_c_ (%)	5.86			7.92			9.35		
**BET**	X_m_ (dry mass)	9.30	7.89	6.37	7.78	6.72	5.80	4.97	6.65	3.86
	C	32.7	24.10	18.09	27.30	20.11	15.11	34.90	25.44	18.92
	M_c_ (%)	5.21	10.21	4.00	12.88	5.04	8.16	7.39	2.92	2.91
	AverageM_c_ (%)	6.48			8.69			4.41		
**Smith**	A	−0.14	−0.164	−0.194	−0.120	−0.139	−0.163	−0.185	−0.173	−0.153
	B	−0.97	−0.954	−0.997	−0.998	−0.714	−0.999	−0.999	−0.854	−0.734
	M_c_ (%)	17.5	23.21	26.66	24.46	26.81	26.12	23.11	20.50	18.59
	AverageM_c_ (%)	22.46			25.80			20.73		
**Oswin**	A	0.16	0.142	0.116	0.137	0.119	0.102	0.107	0.083	0.071
	B	0.33	0.421	0.406	0.427	0.414	0.394	0.458	0.375	0.410
	M_c_ (%)	2.86	7.06	2.13	4.98	3.70	3.27	7.86	1.70	0.18
	AverageM_c_ (%)	4.02			3.98			3.24		
**Peleg**	K_1_	0.20	0.198	0.148	0.193	0.164	0.133	0.149	0.106	0.091
	K_2_	1.44	1.354	1.038	2.383	2.456	2.841	3.979	3.996	4.117
	n_1_	0.43	0.555	0.468	0.572	0.540	0.468	0.588	0.462	0.475
	n_2_	6.816	7.485	6.768	8.413	8.68	8.876	9.074	8.945	9.514
	M_c_ (%)	1.37	2.7	0.25	1.68	0.71	0.48	5.58	1.53	1.01
	AverageM_c_ (%)	1.44			0.96			2.71		

**Table 4 foods-12-02628-t004:** Maximum water-activity values at different temperatures for the stability of the antioxidant powders.

Sample	Temperature		
	20 °C	30 °C	40 °C
Non encapsulated	0.134	0.153	0.175
Encapsulated	0.126	0.147	0.169
Freeze-dried	0.148	0.171	0.193

## Data Availability

The data used to support the findings of this study can be made available by the corresponding author upon request.

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
