# Peer review of "Antioxidant Potential of Spray- and Freeze-Dried Extract from Oregano Processing Wastes, Using an Optimized Ultrasound-Assisted Method"

_foods, 2023, doi:10.3390/foods12132628_

Round 1

Reviewer 1 Report

Introduction

This section should be more balanced. It covers the ultrasound extraction and not much else.

Line 72 Moreover, drying is one of the oldest methods of food preservation, as it reduces excess water.

Please replace “ excess water” with “reduces water activity.”

Methodology:

Line 144: did you remove the ethanol before drying?

Line 148: Please provide more details regarding spray drying. You should at least provide either a value for the feed flow rate or the outlet temperature. Since you discuss the effect of the atomization, please indicate the atomization flow rate as well.

Line 149

What was the solution's initial solids content or brix before adding maltodextrin?

Line 155. Did you keep measuring the sample weight until constant from inside a freeze dryer?

Results and Discussion

Line 214

Did you validate the optimized extraction conditions?

Line 258

Please correct the significant digits of Table 2. You can select two or three.

Line 272

Why this analysis? This makes no sense regarding your work.

Line 352 Please explain your results. Why TPC was lower in the encapsulated powder rather than non-encapsulated? Does maltodextrin protect the phenolics, or doesn’t during spray drying?

Line 360 I don’t understand “reducing power” what do you mean?

Line 365

Please explain better what this efficiency is. Is this the process yield or encapsulation efficiency? How did you calculate this?

Line 292

I don’t understand why this is here. It would be best if you made an effort to make this valuable for your scientific paper.

Line 405

This paragraph isn’t a discussion of the results but rather an introduction. Please discuss your results from figure 5 here

Line 418

Conclusion is a summary of what was done. Please make a proper conclusion regarding your “conclusions” of the work.

Figure 1 is too poor. Please find a way to present this data better. Most of the parameters have no units.

Figure 2 is better but the contour plots should have proper x and y labels

Figure 5 x-axis label should be “Time (days)” ; y-axis lable should not be just some unitsbut the actual measurement.  

Minor improvements in the use of English for better clarification and flow. 

Author Response

Thank you for your comments. Please see the revised manuscript for adopted changes and the attached file for our reply to your comments.

Reviewer 2 Report

The comments for authors:

1. The title of the article should be changed since the antioxidant activity is not in the main focus. The authors should follow the content.

2. Abstract section lacks the conclusions. Please, improve it.

3. What kind of innovative products have been produced in the pharmaceutical, cosmetic and insecticide industries? Please, be more specific.

4. How much waste have been generated from the plant material after the oil separation? Please, include some data.

5. The sentence "Several factors such as time, temperature, the type of solvent, etc. are contributing to the efficiency of the ultrasound extraction." is not clearly written. These factors not always contribute to the extraction efficiency. They may also have negative effect regardless of extraction technique.

6. Please cite the reference for the sentence "To the best of our knowledge, there are few data in the literature on the utilization of distilled oregano wastes.".

7. Introduction: Authors need to clarify what their study brings in novelty in comparison to the similar studies in scientific literature.

8. Section 2.2. The ultrasonic power/amplitude is one of the main parameters that affect the UAE efficiency. Please, add this parameter in the sentence "Various parameters can affect ultrasound extraction efficiency, such as temperature, solvent ratio (S/S), type of solvent, sample pretreatment, and extraction time.".

9. Please, include statistical analysis in Figure 5 in order to discuss significant difference in lipid oxidation.

10. Please, cite the references for the sentence "Several studies utilizing distillation, ultrasound, microwave, or supercritical extraction techniques have highlighted the rich composition in bioactive ingredients present in oregano and the potential for application in the food and pharmaceutical industries.".

11. Line 316-322 - This text should be reduced or removed because it is a repetition of information.

12. Line 323-329 - This text is not important for the discussion part.

13. Line 332-334 - Please include the values of antioxidant activities.

14. The authors should explain the abbreviation just the first time they are mentioned in the text. In the following text, only abbreviations should be used.

15. The authors should improve the discussion part indicating the novelty and originality of their study. 

16. The conclusion section should be improved by explaining the significance of the obtained results and the future perspective.

17. The authors should cite more recent studies.

Author Response

Thank you for your comments and suggestions. Please see attached file for our replies and the revised manuscript that includes all the changes/additions.

Round 2

Reviewer 2 Report

The authors responded to the comments and significantly improved the quality of the manuscript.

Author Response

Thank you very much for your contribution.